# Fabrication of High Surface Area Microporous ZnO from ZnO/Carbon Sacrificial Composite Monolith Template

**DOI:** 10.3390/mi13020335

**Published:** 2022-02-20

**Authors:** Kunal Mondal, Monsur Islam, Srujan Singh, Ashutosh Sharma

**Affiliations:** 1Department of Chemical Engineering, Indian Institute of Technology, Kanpur 208016, Uttar Pradesh, India; srujansingh91@gmail.com; 2Materials Science and Engineering Department, Energy and Environment Science and Technology Directorate, Idaho National Laboratory, Idaho Falls, ID 83415, USA; 3Institute of Microstructure Technology, Karlsruhe Institute of Technology, Hermann-von-Helmholtz-Platz 1, 76344 Eggenstein-Leopoldshafen, Germany; monsur.islam@kit.edu

**Keywords:** ZnO, porous metal oxides, bet surface area, sacrificial template

## Abstract

Fabrication of porous materials from the standard sacrificial template method allows metal oxide nanostructures to be produced and have several applications in energy, filtration and constructing sensing devices. However, the low surface area of these nanostructures is a significant drawback for most applications. Here, we report the synthesis of ZnO/carbon composite monoliths in which carbon is used as a sacrificial template to produce zinc oxide (ZnO) porous nanostructures with a high specific surface area. The synthesized porous oxides of ZnO with a specific surface area of 78 m^2^/g are at least one order of magnitude higher than that of the ZnO nanotubes reported in the literature. The crucial point to achieving this remarkable result was the usage of a novel ZnO/carbon template where the carbon template was removed by simple heating in the air. As a high surface area porous nanostructured ZnO, these synthesized materials can be useful in various applications including catalysis, photocatalysis, separation, sensing, solar energy harvest and Zn-ion battery and as supercapacitors for energy storage.

## 1. Introduction

The development of nanostructures of oxide materials with a high surface area to volume ratio with enough stability is vital for many applications needing improved surface chemistry [1,2,3,4,5,6,7,8,9,10,11,12,13,14]. Porous nanomaterials have gained significant interest and are being studied extensively in this respect [15,16]. The unique microstructures of those porous materials have engrossed considerable attention as a functional material owing to their variety and better performance during applications in the area of catalysis [17], photocatalysis [18], separation [19], adsorption [20], energy storage and conversion [21], and sensing to physicochemical devices [22,23,24] in recent years. These materials are usually characterized for their porosity at macro-, meso-, and microscale, uniformity of pores, hierarchy and interconnection between each level of porosity, large accessible high surface areas, low density and excellent accommodation capability of other nanomaterials into the pore volumes, which enables the easy transport of ion/electron/reactant and the diffusion of mass, exposing boundless significance in applications requiring high accessible surface for reactions such as the energy density, rate capacity and cycling life in energy storage and modulation of electronic activities for sensing [25,26]. Additionally, porous materials can improve the structural stability of the electrode materials for energy storage and sensor devices for their increasing cyclic life, owing to the large porous space and the interconnection of pores at various micro and nanoscopic length scales, which can accommodate not only the volume change [27] but also absorption and dispersion of chemical species [28,29] during a chemical reaction.

For the synthesis of porous nanostructured materials, there are several approaches, including soft and hard templating processes, that have been reported [30,31]. Additionally, electrochemical anodization [27], directed assembly [32], and a self-formation method based on a spontaneous phenomenon, using the chemistry of organic metal alkoxides and alkyl metals, was also strongly considered and effectively applied to the fabrication of a range of metal oxides, and composites of them, aluminophosphates, and silicoaluminate, with hierarchically porous architecture [33,34]. These methods can produce porous materials of various shapes and sizes with controlled pore morphologies.

Among these reported methods, the sacrificial templating process has shown utmost promise to synthesize porous nanomaterials [35,36]. Instances include the use of anodic alumina and polycarbonate templates for the synthesis of various porous metal nanostructures [37]. Some of these approaches use quite expensive sacrificial nano-templates and the removal of these templates is often challenging and overwhelming. Recently, silica nanostructures were used as templates to produce various carbon, metal oxide and metal porous nanostructures [38,39]. Carbon is another low-cost, easily available sacrificial template particularly for the fabrication of various porous nanostructures due to its favorable crystallography, and encouraging sphere/tuber/wire morphology [40]. However, the removal of carbon templates can be challenging. Interestingly, it can be easy to remove carbon templates under high-temperature heat treatment. Such a sacrificial carbon templating method can offer a distinct advantage for the synthesis of various porous/hollow nanostructures in a single step to synthesize various metal/metal oxide nano/microstructures for diverse applications. A thorough understanding of the mechanism of nucleation of nanostructures and removal of template and formation of pores is essential and much needed to control the nature of the resultant porous materials.

Here, using carbon as a sacrificial template, we carried out detailed control experiments to fabricate porous ZnO nanostructures and gain an understanding of the critical parameters for nanopore formation. Using these methods and understanding, we showed that it is possible to synthesize high BET surface area microporous ZnO nanostructures from ZnO/carbon composite monoliths. Synthesized high surface area porous zinc oxides with a specific surface area of ~78.1 m^2^/g that are one order of magnitude greater than that of ZnO nanotubes are reported in the literature. These porous metal oxides were prepared using a novel ZnO/carbon composite architecture where the carbon was used as a template and removed by heating at high temperature in presence of air. We also carried out X-ray Diffraction (XRD), Field Emission Scanning Electron Microscope (FE-SEM) and Transmission Electron Microscope (TEM) studies to investigate the crystallinity, nanomorphology of the ZnO that is critical for applications including catalysis, photocatalysis, separation, sensing, solar energy harvest and Zn-ion battery and as a supercapacitor for energy storage.

## 2. Experimental Section

A simple sol-gel technique was used to prepare the precursor monolith. A total of 734 mg of resorcinol (R, Sigma-Aldrich, St. Louis, MO, USA) and zinc chloride (ZnCl_2_) was added to 4 mL acetone (Fischer Scientific, Waltham, MA, USA). A total of 1.5 mL of formaldehyde (F) was added to this mixture and stirred over a hot plate to ensure proper mixture, which was essential to initiate the sol-gel reaction. The mixture was sealed and kept at room temperature for 12 h to complete the gelation process. The gel was further kept at 60 °C for 12 h in an oven to obtain a dry precursor monolith. The resulting monolith appeared dark red.

The ZnCl_2_/RF monolith was carbonized in a horizontal tube furnace. A typical carbonization recipe was used, as mentioned several times for the fabrication of carbon microelectromechanical (C-MEMS) devices [41,42,43,44,45]. Briefly, the furnace temperature was raised to 900 °C from room temperature with a heating rate of 5 °C/min, followed by a dwell at 900 °C for 2 h. An ambient cooling was implemented after the dwell at 900 °C. A constant nitrogen gas flow was maintained at a flow rate of 0.15 L/min throughout the entire process. These carbonization steps were followed to ensure mechanically stable and completely carbonized monoliths, as observed in the previous study [46]. An RF monolith was also prepared and carbonized for comparison using similar procedure. Upon carbonization, the ZnCl_2_/RF monolith-derived material was calcinated at 600 °C for 1 h using a heating rate of 5°C/min to remove the carbon counterpart of the monolith and obtain porous ZnO nanoparticles.

The surface morphology of the prepared samples was characterized by field emission scanning electron microscopy (FESEM, Quanta 200, Zeiss, Oberkochen, Germany). The composition of the prepared sample was investigated using X-ray diffraction (XRD) using an X’Pert Pro (PAN analytical, Almelo, The Netherlands) X-ray system equipped with Cu K_α_ radiation. We used transmission electron microscopy (TEM, Tecnai G^2^, Hillsboro, OR, USA) for investigation of structural insights of the synthesized material. The porosity of the carbonized and oxidized materials was characterized by performing nitrogen gas adsorption–desorption experiments at 77 K using an ASC-1 setup (Quantachrome Instruments, Boynton Beach, FL, USA). The surface area was calculated based on the gas adsorption–desorption experiments using the Brunauer, Emmett and Teller (BET) model.

## 3. Results

Drying the precursor monolith in a sealed container allowed slow evaporation of the solvent molecules. The addition of ZnCl_2_ was apparent due to the physical appearances of the monoliths. The RF monolith appeared red, whereas the ZnCl_2_/RF monolith had a dark brownish-red color. The appearance change could be attributed to the formation of strong coordination bonds between the Zn^2+^ ions and the phenolic network of RF.

A black monolith was obtained upon carbonization of the ZnCl_2_/RF monolith (Figure 1a), which was transformed into green nanoparticles upon calcination (Figure 1b). We performed Energy-dispersive X-ray spectroscopy (EDX) analysis of the ZnO/Carbon monolith and ZnO obtained after the burning of carbon from the ZnO/Carbon monolith and shown in Figure 1c,d. This analysis was needed to confirm the elemental composition of the monoliths and to confirm if there is any residual carbon in the ZnO after the calcination. It was found that there was no significant carbon present in the obtained ZnO powder.

XRD diffractograms of the materials are presented in Figure 2. The carbonized sample obtained from the RF monolith featured broad peaks around 2Ɵ = 26° and 2Ɵ = 44°, which correspond to the reflections of (002) and (101) planes of carbon [47,48]. These broad peaks are characteristics of the turbostratic microstructure of carbon material, which agrees with previous publications dealing with RF-derived carbon materials [49,50]. In the XRD diffractogram of the carbonized ZnCl_2_/RF monolith, along with the strong reflection of (002) plane of carbon at 2Ɵ = 26°, distinctive peaks of ZnO appeared at 2Ɵ = 31.6°, 34.3°, 36.1°, 47.5° and 56.5°, which correspond to (100), (002), (101), (102) and (110) crystal planes of ZnO and matched with the International Centre for Diffraction Data (ICDD) card number 36-1451. Several peaks of metallic Zn were also present in the diffractogram. The peaks for carbon and metallic Zinc disappeared in the XRD pattern of the calcinated sample; only peaks for ZnO were present in the diffractogram.

The Raman spectra shown in Figure 3 were captured at room temperature. Raman signals are subtle to the crystal structures and the defects associated with ZnO. A major sharp peak was detected at 447 cm^−1^, which is characteristic of hexagonal wurtzite ZnO. This peak corresponds to the Raman active optical phonon mode E2 (high) [51]. Other small peaks were observed at 334.6 cm^−1^ and 584.7 cm^−1^. The broad peak at 1153 cm^−1^ is due to multiple phonon scattering, and the peak at 584.7 cm^−1^ is observed due to structural defects such as oxygen deficiency [52]. The intensity of the peak at 100 cm^−1^ as compared to other peaks is high and corresponds to the Raman active optical phonon mode E2 (low), which indicates high crystal quality, and this result is consistent with the XRD [53].

Figure 4 presents SEM images of the samples prepared here. The RF gel-derived carbon seemed to feature a non-porous morphology, as no visible pores were observed under SEM (Figure 4a). Rather, the microstructure resembled a paste-like morphology, as depicted in Figure 4b. The ZnO/carbon sample featured crystallites with various sizes within the carbon matrix, as shown in Figure 4c,d. In contrast to the carbonized samples, the ZnO material obtained after oxidization of ZnCl_2_/RF-derived carbon monolith featured porous morphology with interconnected ZnO crystallites (Figure 4e). High magnification FESEM revealed that the ZnO crystallites featured a nanorod-like morphology with a diameter ranging from 500 nm to 1 µm. The ZnO material itself featured several tiny pores. The examples of such pores are indicated by the red circles in Figure 5a, where a TEM image of the porous ZnO is presented. The FFT of the TEM image further confirmed the formation of ZnO during the oxidation step.

We characterized the porosity of the ZnO/carbon monolith and the ZnO samples. Figure 6a presents the isotherms of the nitrogen adsorption–desorption of the samples. The isotherm of the ZnO/carbon monolith suggested that the material was mainly meso-porous (2 nm ≤ pore size ≤ 50 nm). The stiff rise of the isotherm observed at the higher pressure also indicated the presence of micropores (pore size < 2 nm). This was further confirmed in the pore size distribution of the ZnO/carbon monolith (Figure 6b), where the majority of pore diameters ranged from 1 Å to 10 nm. The presence of macropores (pore size > 50 nm) up to 1 µm pore diameter was also observed in the pore size distribution. The ZnO/carbon monolith featured a bet surface of 28 m^2^/g, with a total pore volume of 0.001 cc/g. In contrast to the ZnO/carbon monolith, the ZnO sample was mostly micropores, as indicated by the isotherm (Figure 6a) and the pore size distribution (Figure 6b). Even though the pore size ranged up to 300 nm, the pore size of the maximum occurrences was 1.3 nm. The bet surface area and total pore volume of the ZnO sample were 78.1 m^2^/g and 0.005 cc/g, respectively. 

## 4. Discussion

The carbonization of RF gel is studied several times previously for the preparation of porous carbon monolith [54,55,56,57,58]. During carbonization, RF gel goes through a thermo-chemical cleavage process, which results in the loss of volatile gases and leaves a carbon-rich porous material. With increasing temperature, remaining heteroatoms, including oxygen and hydrogen leaves the carbon matrix. In our work, a similar phenomenon also occurred for the precursor monoliths, both RF and ZnCl_2_/RF. For the ZnCl_2_/RF monolith, the RF-derived carbon further reduced the ZnCl_2_ counterpart to metallic zinc (Zn) in a carbothermal reduction mechanism. However, the XRD diffractogram of the carbonized ZnCl_2_/RF (Figure 2) also indicated the formation of ZnO during the carbonization process. Our hypothesis behind such formation of ZnO is that the carbothermal reduced Zn might have reacted to the oxygen-rich volatile by-products generated during the thermochemical cleaving of the RF network to form ZnO nucleation sites. The ZnO nucleation sites further might have initiated lateral crystal growth engulfing local Zn atoms and surrounding oxygen-rich molecules, resulting in larger crystallites, as shown in Figure 4d. The dominant peak of the (101) crystal plane in the XRD diffractogram further supported the fact of lateral crystal growth of ZnO. The localized oxidation of Zn might have also induced localized oxidation of the carbon counterpart, leading to the formation of volatile CO and CO_2_ gases. The escape of these volatile gases further contributed to the formation of the micro- and mesopores within the material, as suggested by the gas adsorption results of ZnO/carbon monolith. Furthermore, the volatile CO and CO_2_ might also have contributed to the ZnO nucleation and growth by locally interacting with the metallic Zn. 

The ZnO obtained from the oxidation of ZnO/carbon monolith appears green, which is unusual compared to the typical white color of ZnO nanomaterials. The coloring can be an important aspect in some applications, e.g., UV sensor. Moreover, in this case, coloring indicates a high density of defects or charged carriers, which will influence the properties and performances of sensors, solar cells, etc. applications. Schulz et al. reported that the color of ZnO crystal varies depending on the concentration of oxygen vacancies within the material [59]. Two phenomena occurred during the oxidation of the ZnO/carbon monolith: the oxidation of unreacted Zn to ZnO and the oxidation of the carbon counterpart to form volatile CO and CO_2_ gases. The vapor pressure of oxygen and the volatile components during the process might have caused the variation in the concentration of the oxygen vacancies, resulting in the green appearance of the obtained ZnO. The escape of the volatile gases from the quite homogenous monolith of ZnO/carbon resulted in the formation of the micro- and mesopores, shown in Figure 4e and evidenced by the gas adsorption presented in Figure 6. The ZnO obtained after the oxidation mostly featured nanorod-like morphology (Figure 4f). The formation of the nanorods might be triggered by the already formed ZnO crystallites in the ZnO/carbon monolith. During the oxidation, the oxidation and growth of the ZnO crystallite occurred in the [1]-direction, which is perpendicular to the hexagonal facet and features the lowest energy for the crystal growth [60]. The preferential growth in the [1]-direction led to the formation of the nanorod shapes. Such direction growth was also supported by a strong (002) peak observed in the XRD diffractogram (Figure 4). Several agglomerated ZnO structures were also observed in the oxidized sample, which might have resulted from the Zn/ZnO embedding within the carbon matrix and the formation during oxidation of the carbon counterpart. Adjacent Zn/ZnO particles were oxidized to ZnO and formed the interconnected network of ZnO, which also contributed to the formation of meso- and macropores. It should be noted that the resulting ZnO featured a very high surface area. Typically, ZnO nanostructures feature a surface area ranging from 3 m^2^/g to 50 m^2^/g [61,62,63,64,65]. The surface area (78.1 m^2^/g) reported here, to the best of the authors’ knowledge, is the highest for ZnO nanostructures reported to date (see Table 1). 

## 5. Conclusions

To conclude, we prepared microporous ZnO nanostructures featuring a high specific surface area. Carbonization of the starting material ZnCl_2_/RF monolith obtained through the sol-gel process resulted in a composite material featuring Zn and ZnO embedded within the carbon matrix. Further oxidation of the carbon composite yielded the formation of ZnO with a mixture of nanorod morphologies and particle agglomeration. The oxidation of the carbon matrix led to the formation of micro- and mesoporous structures of the synthesized ZnO, which yielded a high surface area. Furthermore, synthesis from such carbon-rich material led to the unique green color of the ZnO material. The morphology and high surface area of the resulting green ZnO nanostructures will be highly advantageous in the development of high-performance devices for several applications, including sensors, catalysis and photocatalysts, UV-detectors, energy storage devices and solar cells.

## Figures and Tables

**Figure 1 micromachines-13-00335-f001:**
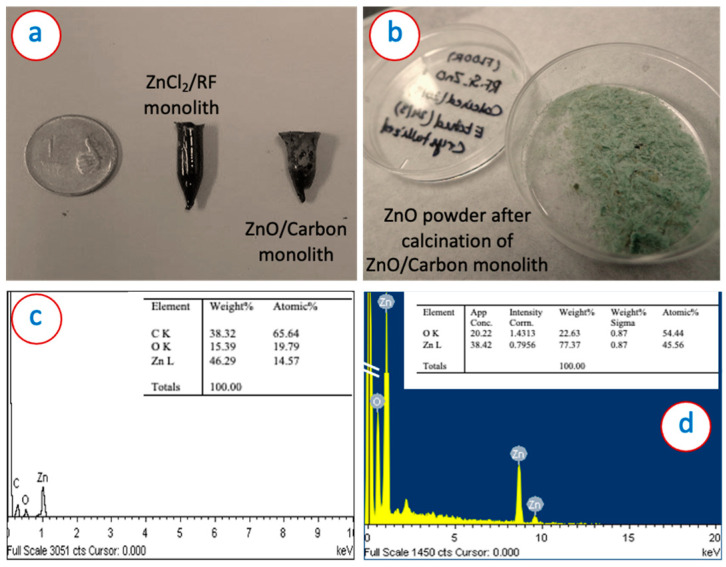
Digital image of (**a**) ZnCl_2_/RF and ZnO/Carbon monolith, and (**b**) ZnO powder obtained from the calcination of ZnO/carbon monolith. EDS spectra and corresponding atomic wt.% of (**c**) ZnO/carbon monolith, and (**d**) ZnO powder obtained from the calcination of ZnO/carbon monolith.

**Figure 2 micromachines-13-00335-f002:**
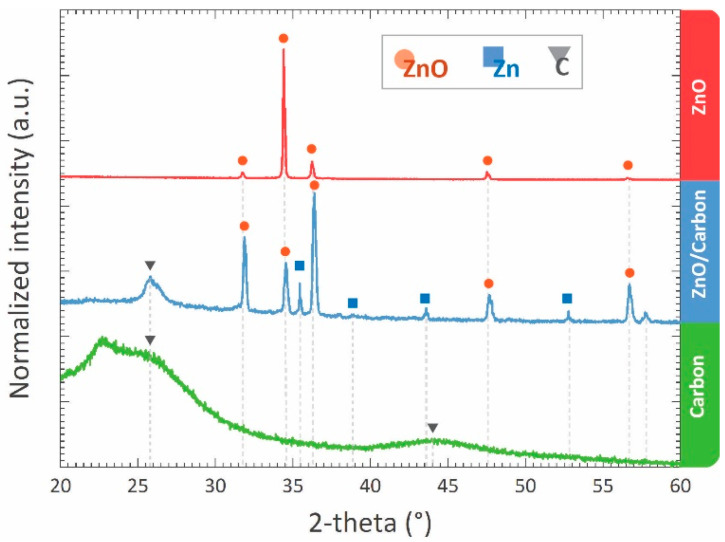
XRD diffractogram of carbonized RF gel, carbonized ZnCl_2_/RF monolith, and the sample obtained after the oxidation of ZnCl_2_/RF-derived carbon monolith.

**Figure 3 micromachines-13-00335-f003:**
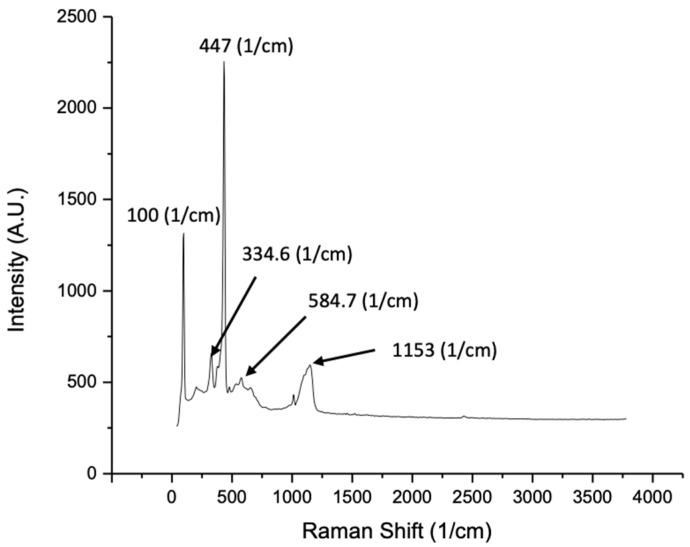
Raman analysis of ZnO powder obtained after calcination of ZnO/carbon monolith.

**Figure 4 micromachines-13-00335-f004:**
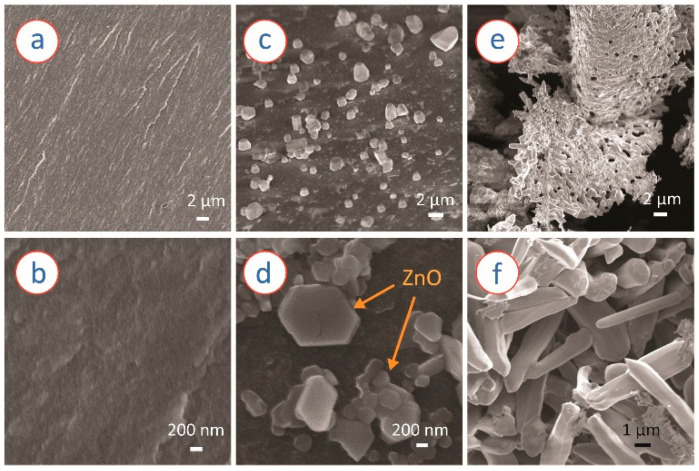
SEM image of (**a**,**b**) RF-derived carbon monolith, (**c**,**d**) ZnCl_2_/RF-derived ZnO/carbon monolith, and (**e**,**f**) ZnO nanostructures obtained from the oxidation of ZnO/carbon monolith.

**Figure 5 micromachines-13-00335-f005:**
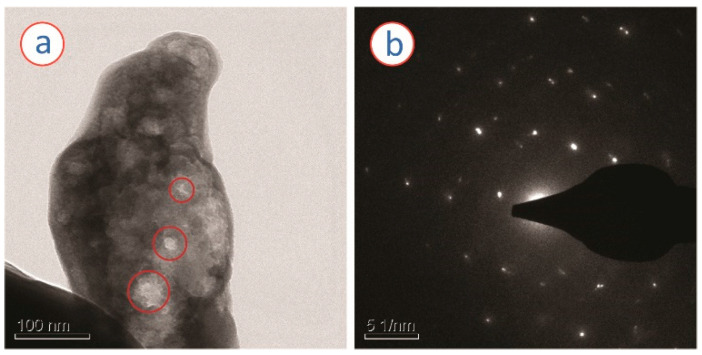
(**a**) TEM and (**b**) SAED of the ZnO nanostructures obtained from the oxidation of ZnO/carbon monolith. The red circles in (**a**) indicate the presence of pores within the material.

**Figure 6 micromachines-13-00335-f006:**
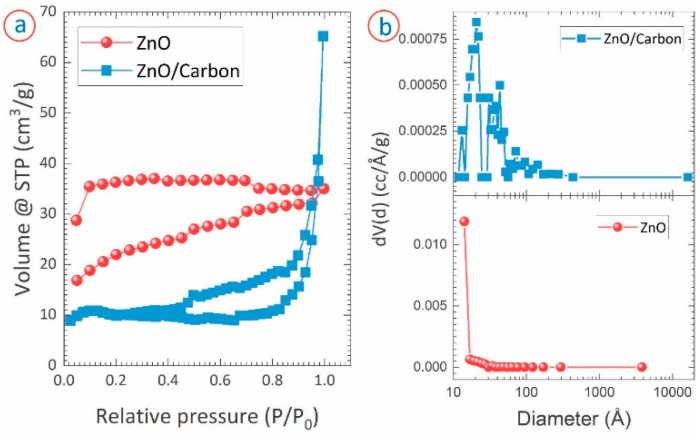
(**a**) Nitrogen adsorption–desorption isotherm and (**b**) pore size distribution of ZnO/carbon monolith obtained from the carbonization of ZnCl_2_/RF and ZnO nanostructures obtained from the oxidation of ZnO/carbon monolith.

**Table 1 micromachines-13-00335-t001:** Summary of surface area of ZnO obtained reported in the literature.

Synthesis Method	Surface Area (m^2^/g)	Average Pore Size (nm)	Method Used for Surface Area	Reference
Precipitating method	25.36	16.0	BET nitrogen adsorption–desorption	[61]
Template method	24	17.4	BET nitrogen adsorption–desorption	[62]
Combustion method	8–22	-	BET nitrogen adsorption–desorption	[63]
Hydrothermal method	10.5	18.6	BET nitrogen adsorption–desorption	[64]
Solvothermal method	10.5	3.5	BET nitrogen adsorption–desorption	[65]
Carbon monolith-based template method	78.1	1.3	Nitrogen adsorption–desorption	This work

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
