# Peer review of "Fabrication of High Surface Area Microporous ZnO from ZnO/Carbon Sacrificial Composite Monolith Template"

_micromachines, 2022, doi:10.3390/mi13020335_

Round 1

Reviewer 1 Report

The paper by Mondal et al. reports the fabrication of microporous ZnO from ZnO/carbon composite monolith. The prepared microporous ZnO has been characterized using various techniques, including X-ray diffraction (XRD), scanning electron microscopy (SEM), transmission electron microscopy (TEM), and BET surface area. The prepared microporous ZnO exhibited higher surface area than ZnO nanotubes. Overall, this work is suitable for publication in this journal after addressing the following comments:

  1. The authors mentioned that the conversion of ZnCl2/RF monolith to porous ZnO nanoparticles at 600 degrees C for 1 h. What heating rate was used? this should be specified in the experimental section.
  2. The authors should show HRTEM images of the microporous ZnO with the lattice spacing.
  3. The isotherm of the ZnO looks strange. I recommend the authors to repeat the measurement and provide the new isotherm and pore size distribution.
  4. A Table comparing the surface area of the microporous ZnO against other reported ZnO nanostructures can be given.
  5. In Page 5, last Paragraph, "oxidarion" should be changed to "oxidation. The spelling in this paper should be double checked carefully.
  6. In the Introduction, some additional references on the applications of ZnO-based nanostructures, such as Chemical Engineering Journal, 425, 130660 (2021); ACS Applied Nano Materials, 3 (9), 8982-8996 (2020); ACS Appl. Energy Mater. 2021, 4, 1, 755–762; Journal of Materials Science, 56,  7348–7356 (2021); and Sensors and Actuators B: Chemical 209, 889-897 (2015) can be included and cited.

Author Response

The paper by Mondal et al. reports the fabrication of microporous ZnO from ZnO/carbon composite monolith. The prepared microporous ZnO has been characterized using various techniques, including X-ray diffraction (XRD), scanning electron microscopy (SEM), transmission electron microscopy (TEM), and BET surface area. The prepared microporous ZnO exhibited higher surface area than ZnO nanotubes. Overall, this work is suitable for publication in this journal after addressing the following comments:

  1. The authors mentioned that the conversion of ZnCl2/RF monolith to porous ZnO nanoparticles at 600 degrees C for 1 h. What heating rate was used? this should be specified in the experimental section.

Response: We thank the referee for his useful comment. We had used a ramp of 5°C/min. We have added this detail in the revised manuscript.

  1. The authors should show HRTEM images of the microporous ZnO with the lattice spacing.

Response: The authors thank the reviewer for the suggestion. However, the purpose of the TEM was to show the internal pores and composition of the material, which the TEM image and the corresponding SAED image satisfy, we believe. We do not have HRTEM image with lattice spacing, at the moment. New TEM study is time consuming at the event of the ongoing COVID-19 pandemic. As we have already provided enough evidence of the composition of the material through XRD, Raman, SAED, and EDX, we think new TEM study for lattice spacing may not be necessary for this study.

  1. The isotherm of the ZnO looks strange. I recommend the authors to repeat the measurement and provide the new isotherm and pore size distribution.

Response:  We thank the referee for the useful suggestions. Please note that we will do an extensive study on the porosity of this materials along with several other metal oxides and will surely consider the isotherm and pore size distribution analysis.

  1. A Table comparing the surface area of the microporous ZnO against other reported ZnO nanostructures can be given.

Response: We thank the reviewer for this suggestion. We have added the following table in the revised manuscript.

Synthesis method

Surface area (m2/g)

Average pore size (nm)

Method used for surface area

Reference

Precipitating method

25.36

16.0

BET nitrogen adsorption-desorption

[61]

Template method

24

17.4

BET nitrogen adsorption-desorption

[62]

Combustion method

8-22

-

BET nitrogen adsorption-desorption

[63]

Hydrothermal method

10.5

18.6

BET nitrogen adsorption-desorption

[64]

Solvothermal method

10.5

3.5

BET nitrogen adsorption-desorption

[65]

Carbon monolith-based template method

78.1

1.3

Nitrogen adsorption- desorption

This work

  1. In Page 5, last Paragraph, "oxidarion" should be changed to "oxidation. The spelling in this paper should be double checked carefully.

Response: We sincerely apologize for the typo. We have corrected.

  1. In the Introduction, some additional references on the applications of ZnO-based nanostructures, such as Chemical Engineering Journal, 425, 130660 (2021); ACS Applied Nano Materials, 3 (9), 8982-8996 (2020); ACS Appl. Energy Mater. 2021, 4, 1, 755–762; Journal of Materials Science, 56,  7348–7356 (2021); and Sensors and Actuators B: Chemical 209, 889-897 (2015) can be included and cited.

Response: We have added the suggested references in the revised manuscript.

Reviewer 2 Report

The structure should depends on the “carbonation” conditions. The authors had better give some explanation how the carbonation conditions were selected, and how the structure will change with different conditions.

The discussion on oxygen vacancy is vague: it is not clearly stated whether oxygen vacancy concentration is larger or smaller than for usual ZnO. Actually the oxygen vacancy concentration should depends on the “calcination” conditions. What is the ambient of the calcination ? Thermodynamically, oxygen vacancy concentration will be determined by oxygen partial pressure.

The authors also need to consider residual carbon in ZnO after the calcination. Chemical analyses (elemental composition evaluation) had better be added; some discussion should at least be added.

The coloring itself can be a serious problem in some applications, e.g., UV sensor. Moreover, coloring indicates high density of defects or charged carriers, which will seriously affect properties of sensors, solar cells. If the authors consider that the fabricated structure is useful for such applications, they must characterize electrical properties also. At least some discussion is needed.

Author Response

  1. The structure should depend on the “carbonation” conditions. The authors had better give some explanation how the carbonation conditions were selected, and how the structure will change with different conditions.

Response: We thank the referee for his useful suggestions. We have added a discussion on the choice of carbonization condition.

  1. The discussion on oxygen vacancy is vague: it is not clearly stated whether oxygen vacancy concentration is larger or smaller than for usual ZnO. Actually the oxygen vacancy concentration should depends on the “calcination” conditions. What is the ambient of the calcination? Thermodynamically, oxygen vacancy concentration will be determined by oxygen partial pressure.

Response: We thank the reviewer for the insight. We performed the calcination in air. The theory of the oxygen vacancy arises from the green color of ZnO nanoparticles. Schulz et al. [Ref. 59 in the manuscript] mentioned that the color of ZnO depends on the oxygen vacancies, and reported green color for their synthesized ZnO nanoparticles, as result of higher oxygen vacancies in their material. This is already discussed in our manuscript, citing their work. We agree that this will need more work to come to a solid conclusion. However, that is out of the scope of our study, presented here. Therefore, we haven’t done extensive work on this.

  1. The authors also need to consider residual carbon in ZnO after the calcination. Chemical analyses (elemental composition evaluation) had better be added; some discussion should at least be added.

Response: We have performed EDX analysis and has been incorporated in the revised manuscript.

  1. The coloring itself can be a serious problem in some applications, e.g., UV sensor. Moreover, coloring indicates high density of defects or charged carriers, which will seriously affect properties of sensors, solar cells. If the authors consider that the fabricated structure is useful for such applications, they must characterize electrical properties also. At least some discussion is needed.

Response: We thank the referee to this useful suggestion. A related discussion has been included in the revised manuscript.

Round 2

Reviewer 2 Report

It seems that sufficient revisions have been made.